

# Adaptive automation: automatically (dis)engaging automation during visually distracted driving

Christopher D.D. Cabrall[1,*], Nico M. Janssen[2,*] and Joost C.F. de Winter[2]

[1] Cognitive Robotics Department, Delft University of Technology, The Netherlands
[2] BioMechanical Engineering Department, Delft University of Technology, The Netherlands
[*] These authors contributed equally to this work.

Corresponding author
Joost C.F. de Winter,
j.c.f.dewinter@tudelft.nl

## ABSTRACT

**Background.** Automated driving is often proposed as a solution to human errors. However, fully automated driving has not yet reached the point where it can be implemented in real traffic. This study focused on adaptively allocating steering control either to the driver or to an automated pilot based on momentary driver distraction measured from an eye tracker.

**Methods.** Participants ($N = 31$) steered a simulated vehicle with a fixed speed, and at specific moments were required to perform a visual secondary task (i.e., changing a CD). Three conditions were tested: (1) Manual driving (Manual), in which participants steered themselves. (2) An automated backup (Backup) condition, consisting of manual steering except during periods of visual distraction, where the driver was backed up by automated steering. (3) A forced manual drive (Forced) condition, consisting of automated steering except during periods of visual distraction, where the driver was forced into manual steering. In all three conditions, the speed of the vehicle was automatically kept at 70 km/h throughout the drive.

**Results.** The Backup condition showed a decrease in mean and maximum absolute lateral error compared to the Manual condition. The Backup condition also showed the lowest self-reported workload ratings and yielded a higher acceptance rating than the Forced condition. The Forced condition showed a higher maximum absolute lateral error than the Backup condition.

**Discussion.** In conclusion, the Backup condition was well accepted, and significantly improved performance when compared to the Manual and Forced conditions. Future research could use a higher level of simulator fidelity and a higher-quality eye-tracker.

## INTRODUCTION

### Automated driving

Over the last couple of decades, researchers have been studying the viability of automated driving for commercial use. However, automation research has not yet reached the point where fully autonomous driving can be implemented with the promise of a perfect system. Current designs of automated driving systems often focus on applying partial,

conditional, or high automation (*SAE International, 2016*), where the human tasks are that of a supervisor. This supervisory role has brought about other human factor issues, including loss of vigilance, varying workload, fatigue, and loss of situation awareness (*Casner, Hutchins & Norman, 2016*; *De Winter et al., 2014*; *Matthews, 2016*; *Parasuraman & Riley, 1997*).

## The potential of adaptive automation

Adaptive automation has been proposed as a solution to maximize human-machine cooperation (e.g., *De Visser & Parasuraman, 2011*; *Hancock, 2007*; *Inagaki, 2003*; *Kaber & Endsley, 2004*; *Parasuraman, 2000*). In adaptive automation, control functions change to a lower or higher level of automation depending on predetermined criteria, such as momentary workload or situation awareness of the human operator. For example, if during the automated execution of a task the human is measured to be inattentive, the algorithm could switch the automation to a lower level or even turn over control entirely to the human operator to engage the human. Alternatively, if high human workload is detected during a manually-executed task, some or all of the control might automatically be switched to the automation.

## Types of adaptive automation

Algorithms that define how and when automation is invoked and terminated differ greatly. *Sheridan & Parasuraman (2005)* (see also *Parasuraman et al., 1992*; *Inagaki, 2003*) describe five types of methods for implementing adaptive automation: (1) critical-event logic, (2) operator performance measurements, (3) modeling, (4) operator physiological measurements, and (5) hybrid methods, combining multiple of these methods.

Physiological measurements offer the advantage that they can be obtained continuously regardless of whether the automation is active or inactive (*Parasuraman et al., 1992*; *Scerbo et al., 2001*). There are different physiological measures that provide information on the human operator state, including heart rate, skin conductance, and eye movements. For this research study, the focus lies on eye movements because they can be measured non-obtrusively and provide specific information regarding where the driver attends to, as opposed to other physiological indexes which provide a more general index of attentional/arousal. How drivers distribute their visual attention is relevant in driving safety research, as the information relevant to driving is likely to be predominantly visual (*Sivak, 1996*).

## Backup automation

Fundamentally, there are two approaches towards adaptive automation using eye movements. The first approach is backup or background automation, which "allows the driver to drive the vehicle, but watches over them in case of trouble" (*Kyriakidis et al., in press*). For example, it is possible to let the driver control the car manually, and invoke automation if the driver is distracted. This approach may be beneficial for safety, as off-road glances are associated with decrements in performance and safety. For example, a naturalistic driving study found significant associations between eyes-off-road time and standard deviation of lateral position (*Peng, Boyle & Hallmark, 2013*).

Backup automation is similar to real-time distraction-mitigation feedback which alerts drivers based on their off-road eye glances (*Donmez, Boyle & Lee, 2007*). However, alerts alone may not always be effective, as drivers may decide to ignore warning systems (e.g., *Parasuraman & Riley, 1997*).

## Forced manual driving

The second and opposite approach ('foreground automation'; *Kyriakidis et al., in press*) would be to let the car drive automatically, and force the driver to take over if he or she is distracted (i.e., negligent in their responsibility for monitoring the dynamic driving task).

The notion of forced manual driving might seem odd due to its apparent unsafe nature. However, it is not odd in the sense that it roughly corresponds to a path being followed by the automotive industry. As a result of an investigation into the first fatal crash with Tesla's Autopilot and a truck in May 2016, the National Transportation Safety Board (NTSB) has issued recommendations to ''develop applications to more effectively sense the driver's level of engagement'' and to ''incorporate system safeguards that limit the use of automated vehicle control systems to those conditions for which they were designed'' (*NTSB, 2017*). Other than a warning based on hands-on-wheel sensing, one such safeguard could be to automatically activate a functional transition from the automated mode towards manual driving, see the case of Cadillac Super Cruise, which uses head tracking software that ''helps make sure your eyes are on the road, and alerts you when you need to pay more attention or take back control'' (*Cadillac, 2018*). In current level 2 automated driving, the car performs lateral and longitudinal control, and the system penalizes the inattentive supervisor with a transition of control back to the driver. In an overview of 2017 models from vehicle manufacturers with level 2 driving automation systems, transitions of control back to the driver were found to be a commonly employed strategy for reacting to insufficient supervisory driver attention (C Cabrall, A Eriksson, F Dreger, R Happee & JCF De Winter, 2018, unpublished data). Accordingly, the forced manual driving may be a useful strategy to prevent overreliance on automation.

## The present study

In summary, transitions in adaptive automation could occur in two directions. While driving manually, detection of visual distraction could trigger a transition from manual driving control to automated control (Backup automation). In the other direction, visual distraction could trigger a transition from automated to manual driving (Forced manual driving). At present, it is unknown whether background automation or foreground automation with forced manual driving is preferred in terms of safety and driver acceptance.

The present experiment was performed with three different conditions (1) Manual driving (Manual), (2) An automated backup (Backup) condition, consisting of manual driving except during periods of visual distraction, where the driver was backed up by an automated pilot that was automatically initiated, and (3) A forced manual drive (Forced) condition, consisting of automated driving except during periods of visual distraction, where the driver was forced back into the manual control loop.

An expected result was that the automated backup condition would yield better lane-keeping performance during visual distraction because the automation is programmed to

keep lane center better than what humans are capable of. Additionally, it was of interest to see whether people accepted this condition, in which control was taken away from them. For the forced manual drive condition, it was expected that lateral driving performance would deteriorate as compared to the manual drive condition during such moments because visual attention is a prerequisite for being able to keep the car in the lane (_Senders et al., 1967_).

## METHODS

### Ethics statement

This research was approved by the Human Research Ethics Committee (HREC) of the Delft University of Technology (TU Delft). All participants provided written informed consent.

### Participants

Thirty-one people participated, of which 25 were male and six female. The mean age was 26.4 years ($SD = 4.5$ years). Participation criteria were having a driver's license, and not having to wear glasses to see properly. Participants were offered €5 compensation for their time (approx. 30 min).

### Equipment

A SmartEye DR120 remote eye tracker was used to record the participant's gaze direction while seated and viewing a desktop monitor (Fig. 1). Data were collected at a frequency of 60 Hz. The experiment took place in a room with standard office lighting and lowered window blinds. A 24-inch monitor was used to display the simulated environment. The distance between the monitor and the participant differed between participants but was limited by the DR120 eye tracker, which was able to measure in the range 50–80 cm from the cameras. A Logitech G27 steering wheel was used to control the simulated vehicle. PreScan software (TASS International, Helmond, The Netherlands) was used to create the simulation environment. MATLAB/Simulink was used along with PreScan to control the simulated vehicle and to log data. A stack of CDs and a small boom box to the right of the monitor and steering wheel were used to present a secondary task that evokes visual distraction similar to that which might commonly occur while driving (e.g., using a route navigation device, tuning the radio, texting).

### Simulated environment

The environment consisted of a two-lane road with a lane width of 5 m. The road had five straight segments and four 10° bends (Fig. 2). The participant was shown the dashboard of a vehicle (BMW X5) as well as the road in front of them (Fig. 3). A bar on the dashboard indicated the state of the automation. A green bar indicated that the automation was on, a yellow bar indicated that the automation was still on but that the participant was about to regain lateral control, and a red bar indicated that the automation was off (i.e., manual lateral control). The automation was designed in such a way that when it was switched on, it would quickly drive the car towards the center of the right lane and keep it there.

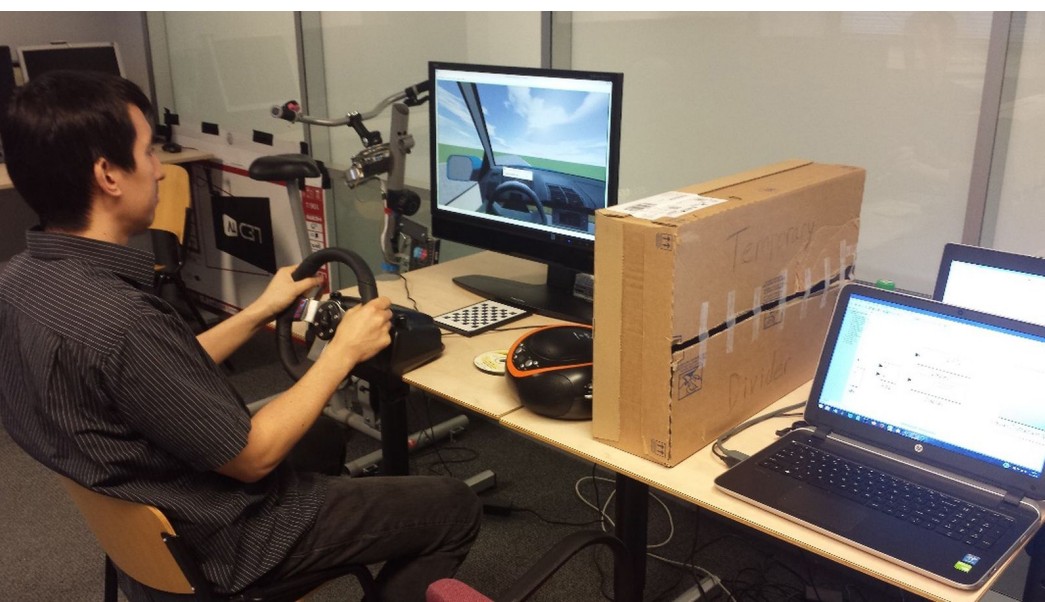

**Figure 1** **The experimental setup.** Source credit: the authors.

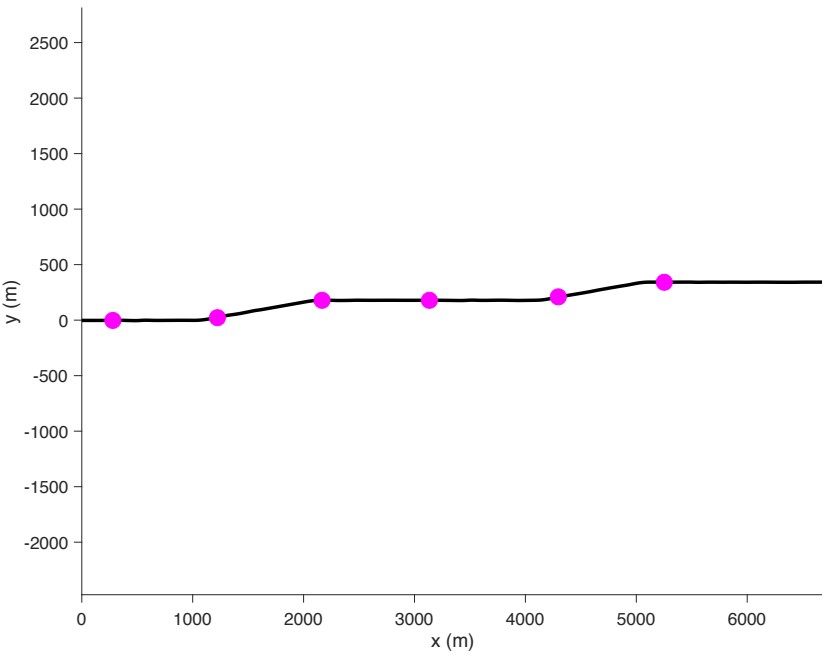

**Figure 2** **Top-down perspective of the road. The markers indicate the six moments when a 1-s beep was presented, signaling that the participant could start the secondary task.**

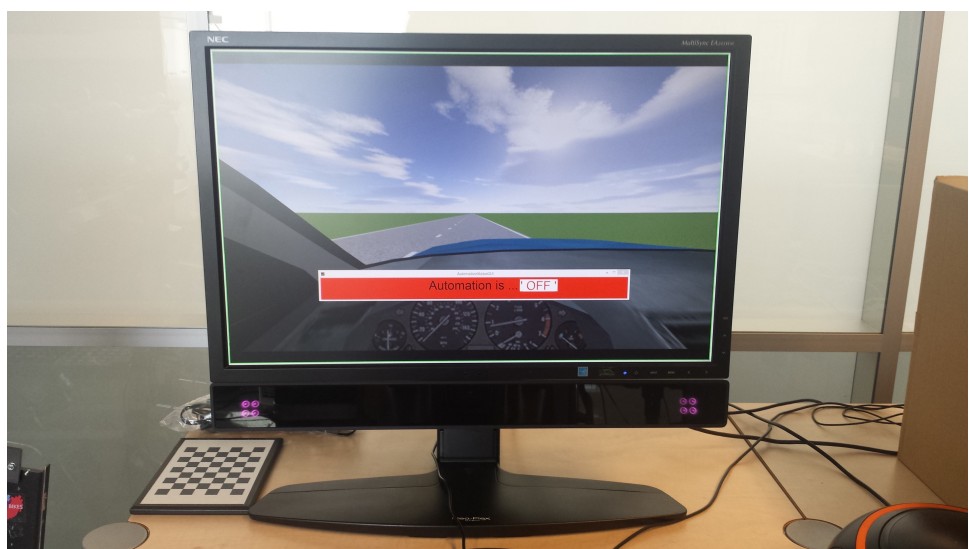

**Figure 3** **Photo from the participant's perspective.** The eye-tracker cameras are connected to the bottom of the monitor. Source credit: the authors.

## Experimental conditions

A within-subject design was used, and the order of the conditions was counterbalanced across the participants. The counterbalancing was done by presenting the six possible orders of the Manual (1), Backup (2), and Forced (3) conditions, in the following manner to the first six participants: 1-2-3, 3-2-1, 2-3-1, 2-1-3, 1-3-2, 3-1-2. These orders were repeated for Participants 7–12, 13–18, 19–24, and 25–30, and Participant 31 was presented with the 1-2-3 order. During the entire experiment, the vehicle speed was constant at 70 km/h, and thus no longitudinal control actions were required. This speed was chosen to simulate driving on rural roads. No infrastructure (buildings, signage, vegetation) nor any other traffic were simulated. Three experimental conditions were used:

Manual condition (Manual). In the Manual condition, the participant performed the steering without help from an automated system.

Automated backup condition (Backup). In this condition, the automated system assumed lateral control when visual distraction was measured. Otherwise, the participant performed manual steering. Visual distraction was defined by the consecutive eyes-off-monitor time being greater than 1.5 s. The secondary task was placed to the right of the steering wheel, and when the participant turned the head to look at it, it sometimes became difficult for the eye tracker to record the eyes. When the eye tracker was not able to record the eyes, it reported this as null values, and the algorithm treated these as off-monitor measurements. Automation termination was also performed based on eye measurements: the participant would regain lateral control if (s)he focused on the monitor for 4.5 s. The yellow status bar switched on 1.4 s before the transition to manual took place. The 1.5 s and 4.5 s thresholds were based on pilot studies (see https://doi.org/10.4121/uuid:49d87edc-07a6-4f07-a5e6-0b699705881b). The 1.5 s

threshold for Backup automation is in approximate agreement with the literature, which suggests that off-road glances of 2.0 s and longer are risky (*Klauer et al., 2006*; *Ryu, Sihn & Yu, 2013*). Recently, *Liang, Lee & Horrey (2014)* concluded that "frequent off-road glances longer than 1.7 s present a high-risk glance pattern in the seconds preceding a safety-critical event and that the 2.0 second-threshold that is frequently cited in defining dangerously long off-road glances might be a liberal estimation".

Forced manual drive condition (Forced). The Forced condition can be described as being opposite to the Backup condition in the sense of control transition directionality. The automation had lateral control of the car while the participant was assessed as being visually attentive, and initiated a control transition to manual driving if visual distraction was measured. If the gaze was directed away from the monitor for 1.5 consecutive seconds, the automation switched off, and the participant would be forced to drive manually. The status bar switched from green to yellow 0.75 before the transition to manual would take place. The algorithm would wait until 4.5 on-monitor seconds were measured and then the automation would switch on.

In the Manual and Backup conditions, the first 3.5 s of each trial were driven with automation enabled, and between 3.5 and 5 s, the status bar was yellow. This ensured that the participant started smoothly with zero lateral error.

During automated driving, the steering wheel (i.e., the physical angle of the Logitech steering wheel) was decoupled from the simulated steering angle, and so not necessarily centered. When regaining manual control, the virtual steering angle would make a discrete jump from the previous steering angle determined by the automation towards the steering angle at which the physical steering wheel angled at that moment.

It is noted that the above-mentioned descriptions of the Backup and Forced conditions are simplifications of the actual algorithms (see https://doi.org/10.4121/uuid:49d87edc-07a6-4f07-a5e6-0b699705881b for source code). One detail is that, to prevent effects of eye blinks and rapid glances between the secondary task and the monitor, the algorithms featured a filter regarding the transition back to the nominal state (i.e., manual driving in the Backup condition and automated driving in the Forced condition). This means that the driver did not have to look to the monitor for 4.5 s consecutively to induce a transition. Specifically, the algorithm of the Backup condition was programmed in such a way that if the eye tracker measured 1.5 consecutive off-monitor seconds, the on-monitor counter would reset to zero. In other words, if a cumulative total of 4.5 on-monitor seconds were measured (i.e., without 1.5 consecutive off-monitor seconds in between), the participant automatically regained lateral control. For the Forced condition, on the other hand, the on-monitor counter would reset after 0.33 consecutive off-monitor seconds. There was no specific purpose for these differences between the Backup and Forced conditions, but these differences were the consequence of adjustments during pilot testing.

## Secondary task

The participant was given a secondary task intended to cause a visual distraction. In this secondary task, the participant was required to perform a sequence of physical actions

involving the stack of CDs and a CD-player (see *Horberry et al., 2006*), who reported that this type of task degrades driving performance).

The sequence of steps consisted of keeping the left hand on the steering wheel and using the right hand to (1) press stop on the CD-player, (2) open the CD-player, take out the CD, and put it on top of the stack of CDs, (3) take out the bottom CD from the stack, put it in the CD-player, and close the lid, (4) press play on the CD-player, (5) put the stack of CDs back in their original position, and (6) place the right hand back on the steering wheel. The sequence of steps was designed to encourage visual distraction and thus trigger an automatic transition of control. Note that the volume of the CD-player was set to zero.

The participant was told to keep the left hand on the steering wheel at all times. Furthermore, the participant was instructed to look at the secondary task (CDs, CD-player) when performing the secondary task. In other words, the participant was not supposed to look towards the monitor and simultaneously perform the secondary task based on peripheral vision or touch. This requirement was included to ensure that the participant was visually distracted from the driving due to performing the secondary task.

At six moments during the drive (after 15 s, 65 s, 115 s, 165 s, 225 s, and 275 s), the participant was alerted that he/she was required to perform the secondary task by a long (1 s) beep. To encourage secondary task engagement, the participant was scored by the experimenter on a scale from 0 to 10. The participant could get up to 6 points for performing the task steps correctly and up to 4 points depending on how quickly the task was completed. The scoring was done by the experimenter by looking at the participant and an on-screen timer that was visible on the experimenter's computer. The precise scoring criteria are provided in https://doi.org/10.4121/uuid:49d87edc-07a6-4f07-a5e6-0b699705881b. If the task was not completed within 25 s, the participant would only get points for the steps finished at that time. The total score was the average of the six secondary tasks per driving trial. At the end of each driving trial and before they started with the questionnaires, the experimenter orally told the participant what the secondary task score was, rounded to 1 decimal point.

Additionally, for the Manual and Backup conditions, a short (0.25 s) beep was produced 25 s after the long beep, to mark the end of the secondary task period. In the Forced condition, the short beep was produced when the automation had made a transition from manual to automated driving or 25 s after the long beep (whichever came first). For the Forced condition, the short beep was presented right after the manual-to-automation transition to signal to the participant that the secondary task was over.

The instructions form mentioned that "a long beep will indicate the start of the task, a short beep will indicate that you can stop the task if you are not already finished." Furthermore, the form stated that lane keeping was the primary task, "Your primary task is to focus on staying in the center of the right lane as accurately as you can. This should always be the most important task. Safety first!". The form also clarified that changing the CD was the secondary task, and that the participant should attempt to score as high as possible while still driving safely.

## Procedure

After reading and signing the consent form, which mentioned the goal of the experiment and the workings of the three conditions, each participant was asked to fill out a personal information questionnaire. They were also required to read the instructions form (see https://doi.org/10.4121/uuid:49d87edc-07a6-4f07-a5e6-0b699705881b).

Next, the participant was asked to sit in front of the eye tracker and focus on four fixed points on the monitor to perform a gaze calibration. If the calibration could not be completed, the participant was asked to sit differently so that the cameras could record their eyes better before performing another calibration.

For each of the three conditions, the participant was asked to drive the simulated vehicle in the environment described above, using the steering wheel for lateral control. Additionally, for each of the three conditions, at fixed intervals during driving, the participant was required to perform the CD-player secondary task.

After each driving trial, the participant was asked to complete a NASA Task Load Index (TLX) questionnaire (*Hart & Staveland, 1988*). Following the Backup and Forced conditions, the participant was required to fill out an acceptance scale of in-vehicle technology (*Van der Laan, Heino & De Waard, 1997*). The participants were not required to complete this questionnaire for the Manual condition, because the scale asks to rate a specific vehicle technology. At the end of the experiment, the participant was asked to complete a questionnaire where they could state which session they preferred as well as give general comments (see https://doi.org/10.4121/uuid:49d87edc-07a6-4f07-a5e6-0b699705881b for all the questionnaires used in this study).

The participant performed a 185 s training run before each of the driving trials to become familiar with each condition. These training runs were driven on the same track as the actual experimental runs, and included three secondary task periods. After the training run, the participant drove the full track, which took 350 s for each driving trial and included six secondary task periods.

## Dependent variables

The following measures and measurements were assessed across the 10.0 s and 349.5 s of elapsed time per driving trial of a particular condition. The first 10 s were discarded because this period was regarded as settling time for participants.
Lateral performance:

- Mean Absolute Lateral Error (meanALE) (m). This was the mean of the absolute difference in lateral position between the vehicle's position and the lane center. The meanALE is an index of *overall* lane keeping performance and includes both periods where the lateral driving automation is active (and so the lateral error is 0) and periods of manual driving.
- Mean Absolute Lateral Error during Manual Driving (meanMALE) (m). This was the mean of the absolute difference in lateral position between the vehicle's position and the lane center, only for moments when the participant was driving manually.

- Maximum Absolute Lateral Error (maxALE) (m). maxALE is the *maximum* of the absolute difference in lateral position between the vehicle's position and the lane center in meters, and can be regarded as an index of safety.

Furthermore, the following measures were extracted from the self-reports, for each of the three driving conditions:

Secondary task performance:

- The secondary task score (0–10) was computed as the mean of the full set of six secondary tasks of a driving trial.

Workload:

- NASA-TLX (%), ranging from 0% to 100% with steps of 5%. This questionnaire was used to assess subjective workload on six different categories: (1) Mental demand, (2) Physical demand, (3) Temporal demand, (4) Performance, (5) Effort, and (6) Frustration (*Hart & Staveland, 1988*). The items were answered on a 21-point scale ranging from 'very low' ('perfect' for the performance item) to 'very high' ('failure' for the performance item). A composite score was obtained by taking the mean of the six different sub-category scores (*Byers, Bittner Jr & Hill, 1989*).

System acceptance:

- Acceptance scale, ranging between +2 and −2, with steps of 1. The acceptance scale was used to assess the drivers' opinion on the Usefulness and the Satisfaction of the systems they tested. This questionnaire consisted of nine sub-scale items, presented in order as (1) useful-useless, (2) pleasant-unpleasant, (3) bad-good, (4) nice-annoying, (5) effective-superfluous, (6) irritating-likeable, (7) assisting-worthless, (8) undesirable-desirable, (9) raising alertness-sleep inducing.
- Preference. The participant was also asked which condition they preferred the most in a final questionnaire after they had performed all of the conditions. The question they were asked was "Which session did you prefer?". The possible answers were "session 1", "session 2" "session 3", and "no difference".

## Statistical analyses

Non-parametric tests were used because some of the performance measures were non-normally distributed among participants. For example, maxALE represents the maximal deviation during the entire drive and so is sensitive to a single road excursion. Differences between pairs of conditions were compared using the Wilcoxon signed rank test. Corresponding effect sizes were calculated as $Z/N^{0.5}$. A significance level of .005 was used (*Benjamin et al., 2017*).

## RESULTS

### Automation functionality

First, we assessed whether the Backup and Forced conditions worked as intended. Figure 4 shows the proportion of participants with automation on at any time for the Backup and Forced conditions. It can be seen that about 90% of the participants in the Backup condition drove automatically about 10 s after the task initiation beep was presented.

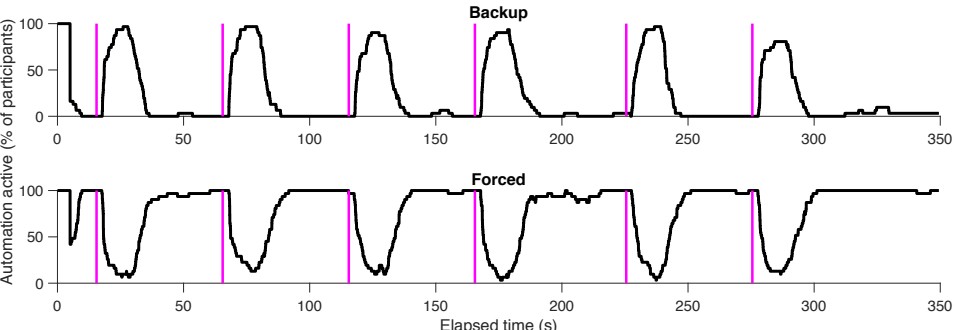

**Figure 4  The proportion of participants with automation turned on as a function of elapsed time.** The magenta vertical lines represent the secondary task initiation beeps. The overall percentage of automated driving time was 0%, 22%, and 76% for the Manual, Backup, and Forced conditions, respectively.

The 10 s comprises the minimum 1.5 s required to initiate a transition, plus individual differences in eye-response time (or the fact that participants may have used frequent scanning back and forth scanning rather than a direct re-allocation of gaze in a binary manner). Similarly, about 90% of the participants were issued manual driving control status in the Forced condition about 10 s after the beep. Figure 4 also shows that some of the participants experienced control transitions outside of the secondary task periods. This could be due to eye tracker imperfections, as faulty measurements could result in 1.5 s off monitor glancing. Summarizing, the results in Fig. 4 show that the Backup and Forced conditions worked in opposite ways, as intended.

## Lane-keeping performance

Figure 5 shows results of the absolute lateral errors for every participant, and of all participants averaged. Differences between conditions are evident in the lateral position while performing a secondary task (i.e., up to about 20 s following each magenta line, cf. Fig. 4). In the Backup condition, the absolute lateral error drops to near-zero after participants were notified to perform the secondary task. In the Manual condition, however, the absolute lateral error increases with evidently higher peak values compared to periods without the secondary task. During the Forced condition, the absolute lateral error is near-zero before the secondary task periods (i.e., when automation is on) but increases substantially when the automation is disengaged.

Figure 6 shows the results for the three lane-keeping performance measures. Concerning the first measure (meanALE), the Backup condition yielded better lane-keeping performance than the Manual condition. Specifically, the meanALE of the Manual condition (Med = 0.54 m, IQR = 0.25 m) was higher than for the Backup condition (Med = 0.33 m, IQR = 0.12 m), $Z = 4.62$, $r = .83$, $p < .001$. The meanALE of the Forced condition (Med = 0.18 m, IQR = 0.20 m) was significantly lower than that of both the Manual condition ($Z = 4.78$, $r = .86$, $p < .001$) and the Backup condition ($Z = 3.02$, $r = .54$, $p = .003$).

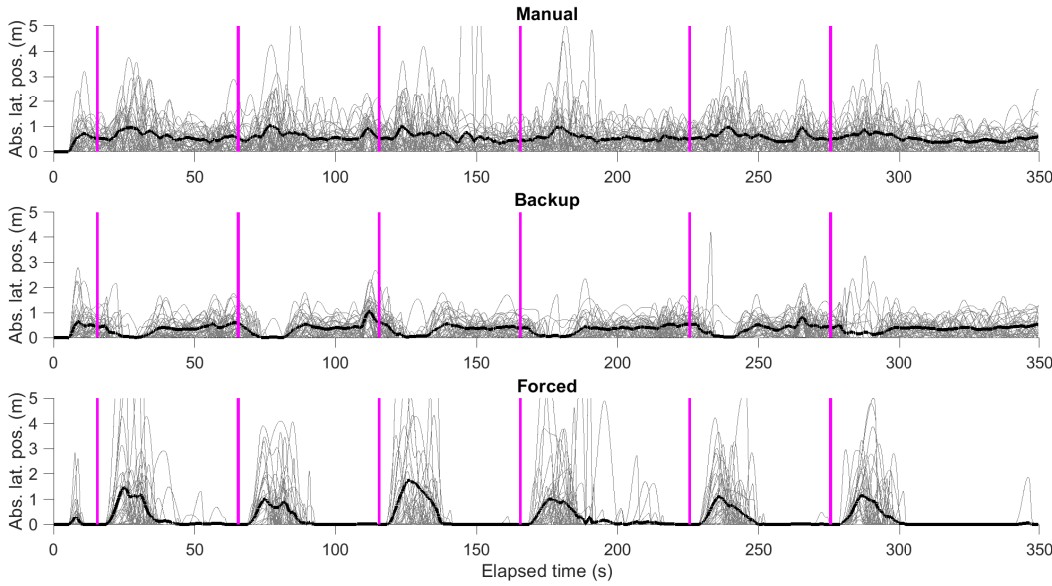

**Figure 5 Absolute lateral position as a function of elapsed time.** The magenta vertical lines represent the secondary task initiation beeps. The results of individual participants ($N = 31$, in each condition) are shown in gray. The mean of participants is shown in black.

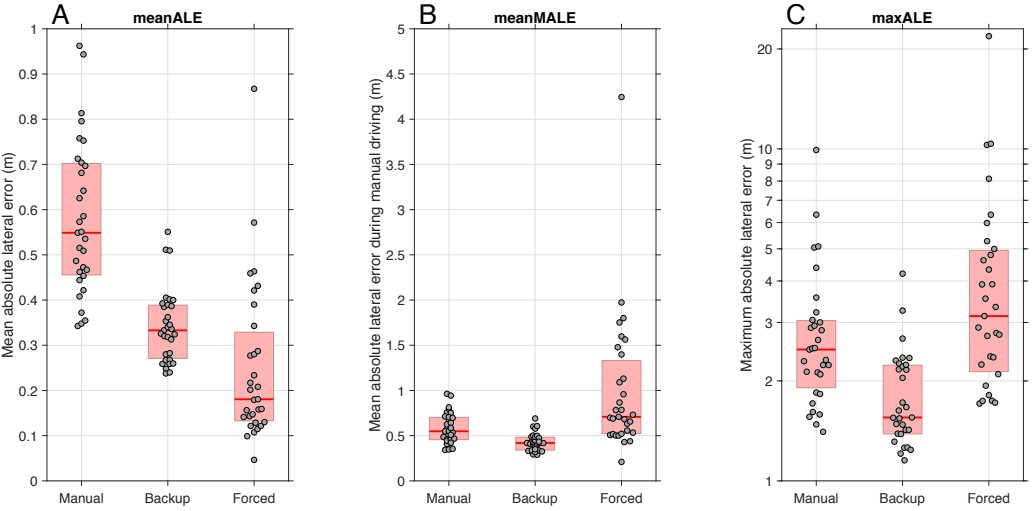

**Figure 6 (A) The mean absolute lateral position (meanALE), (B) the mean absolute lateral position during manual driving (meanMALE), and (C) the maximum absolute lateral position (maxALE).** maxALE is presented on a logarithmic scale. For each box, thick red the horizontal line is the median, and the edges of the box are the 25th and 75th percentiles. The markers represent scores for individual participants, with a horizontal offset to prevent overlap.

Concerning the second measure (meanMALE), which compares only the portions of manual driving, the median value for the Backup condition was 0.42 m (IQR = 0.14 m), which was significantly lower than the Manual condition (Med = 0.55 m, IQR = 0.25 m), $Z = 3.94$, $r = .71$, $p < .001$. The Forced condition yielded a significantly higher meanMALE (median = 0.71 m, IQR = 0.81 m) than the Manual condition ($Z = 3.88$, $r = .70$, $p < .001$) and the Backup condition ($Z = 4.66$, $r = .84$, $p < .001$). In summary, average lane positioning during periods of manual control with adaptive transitions of control was improved in the Backup condition compared to full manual control and was worsened in the Forced condition.

Finally, concerning the third measure (maxALE), the Manual condition yielded poorer performance (Med = 2.49 m, IQR = 1.74 m) than the Backup condition (Med = 1.67 m, IQR = 0.70 m), $Z = 3.51$, $r = .63$, $p < .001$. Furthermore, the maxALE of the Forced condition (Med = 3.14 m, IQR = 2.67) was significantly higher than that of the Backup condition, $Z = 4.23$, $r = .76$, $p < .001$, whereas the difference in maxALE between the Forced and Manual conditions was not statistically significant, $Z = 2.02$, $r = .36$, $p = .044$. In summary, maximum lane deviations were lowest in the Backup condition.

### Driver attention and secondary task performance

Figure 7 shows the percentage of participants glancing at the monitor as a function of elapsed time for the three conditions. It can be seen that participants in the Backup condition were more likely to look away from the monitor (between about 3 and 12 s after the task initiation beep) than participants in the other two conditions. Participants apparently used the available backup to concentrate on the secondary task, whereas in the Manual and Forced conditions, participants had to periodically check the road to keep the vehicle in the lane. This was also reflected in the average number of points earned across the six sessions, with median values of 8.50, 9.00, and 8.67 on the scale from 0 to 10, for the Manual, Backup, and Forced conditions, respectively (Fig. 8). The score for Backup was significantly higher than for the Manual ($Z = 3.02$, $r = .54$, $p = .003$) and Forced condition ($Z = 3.23$, $r = .58$, $p = .001$). The difference between the Manual and Forced conditions was not significant ($Z = 0.58$, $r = .10$, $p = .562$).

### Self-reported workload

The results of the NASA-TLX questionnaires per item are shown in Fig. 9. Generally, the Backup condition yielded lower workload ratings than the Manual and Forced conditions for each of the six items. Regarding composite workload (i.e., the mean across the six items), the medians across participants for Manual, Backup, and Forced were 46.7%, 31.7%, and 46.7%, respectively. The composite workload of the Backup condition was significantly lower than both the Manual condition ($Z = 3.98$, $r = .71$, $p < .001$) and the Forced condition ($Z = 3.95$, $r = .71$, $p < .001$). The difference between the Forced and Manual conditions was not statistically significant, $Z = 1.09$, $r = .20$, $p = .275$.

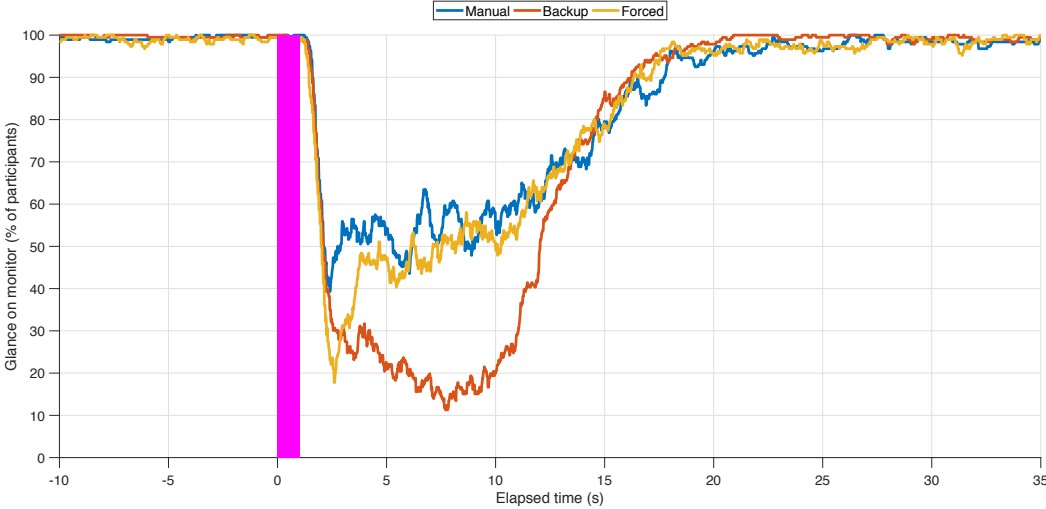

**Figure 7** **The percentage of participants glancing at the monitor as a function of elapsed time.** Filtering with an interval of 0.25 s was applied, and the data for the six secondary tasks were averaged. Missing data (e.g., the eye tracker not tracking the eyes because the participant is performing a blink or performing the secondary task) were coded as an off-monitor glance. The thick magenta vertical line represents the secondary task initiation beep.

## Self-reported driver acceptance

The results of the acceptance scale per item are shown in Fig. 10. Participants reported significantly higher acceptance scores on all items ($p < .001$) for the Backup condition as compared to the Forced condition, except for the *Raising alertness –Sleep-inducing* item.

At the end of the experiment, each participant completed a form where they were asked which session they liked most. Out of the 31 participants, 22 (71%) selected the Backup condition as their preferred condition, eight (26%) selected the Manual condition, and one (3%) participant selected the Forced condition. A final optional comments section was provided through which 13 participants provided responses (see https://doi.org/10.4121/uuid:49d87edc-07a6-4f07-a5e6-0b699705881b ). Three participants reported that they would prefer to change control manually. Furthermore, three participants commented on the automation status bar, which was perceived as annoying, useless, and/or interfering with the working of the systems.

## DISCUSSION

This research aimed to design and investigate a distraction-mitigation system that automatically invoked a control transition based on distraction measurements and to see how it would affect performance, workload, and acceptance. Triggers were designed and implemented under two essentially opposite approaches to eye-based adaptive driving automation to examine different directional consequences upon detection of distraction: a transition from manual to automated control vs. a transition from automated to manual control.

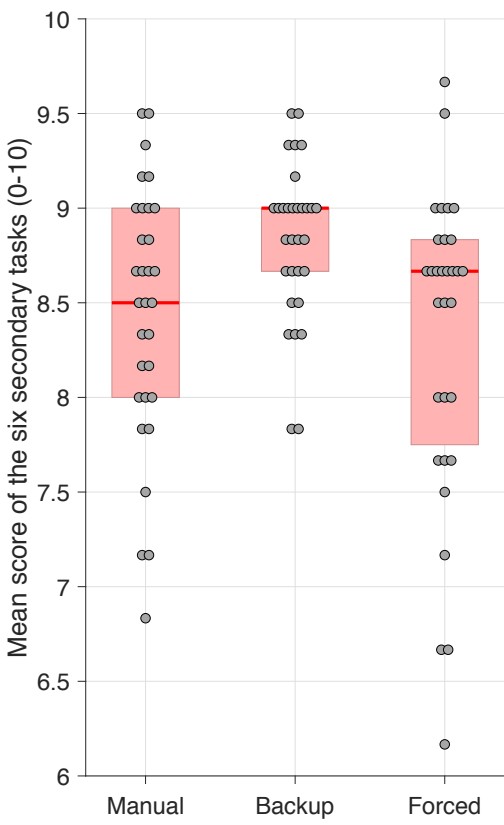

**Figure 8** **Points scored on the secondary task.** The participant's score is the average of six tasks. For each box, thick red the horizontal line is the median, and the edges of the box are the 25th and 75th percentiles. The markers represent scores for individual participants, with a horizontal offset to prevent overlap.

## Lane-keeping performance
### Backup vs. manual

Lane-keeping performance was assessed via three complementary measures: meanALE, meanMALE, and maxALE. All three performance indices were significantly better for the Backup condition compared to the Manual condition. The substantially lower meanALE and maxALE are a direct result of the secondary task that induced visual distraction and triggered the lane centring driving automation. For the meanMALE measure, the enhanced lateral driving performance during periods of manual driving could be explained by a 'staging' benefit in the sense that the automated agent positioned the car in the center of the lane before returning manual control to the driver. However, it could also be because drivers felt more at ease and confident during manual driving, knowing that they had an automated driving agent to support them.

### Forced vs. manual

Regarding the Forced condition, improved lane-keeping performance compared to the Manual condition was found only for the overall performance of meanALE, whereas a performance detriment was found for meanMALE. The superior meanALE of the Forced

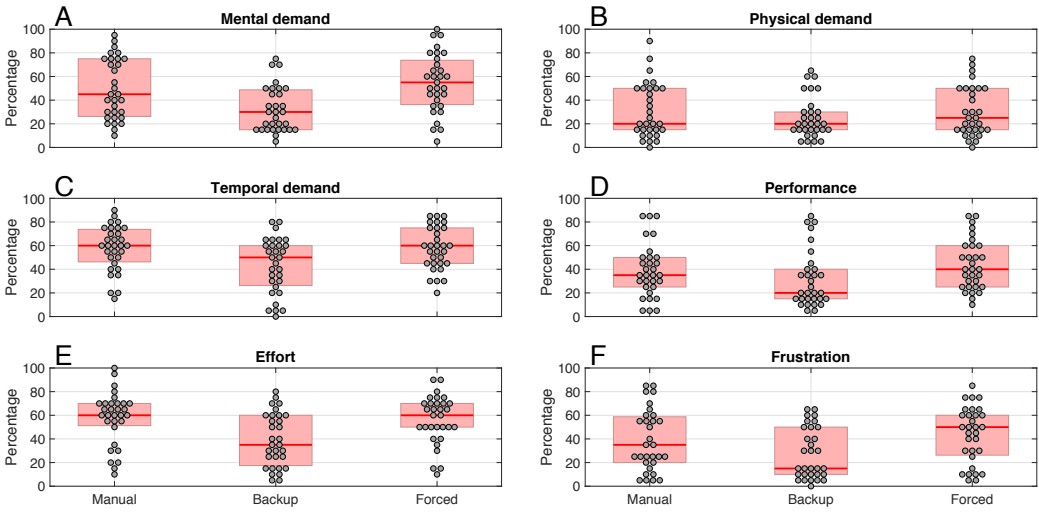

**Figure 9** **Results for the six items of the self-reported workload (NASA-TLX) per condition.** For each box, thick red the horizontal line is the median, and the edges of the box are the 25th and 75th percentiles. The markers represent scores for individual participants, with a horizontal offset to prevent overlap. The items were answered on scale ranging from 0 = 'very low' ('perfect' for the performance item) to 100 = 'very high' ('failure' for the performance item).

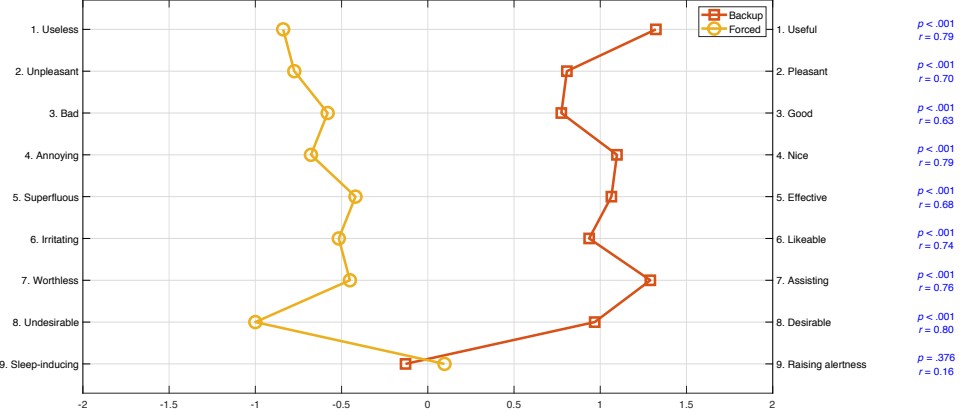

**Figure 10** **Mean ratings on the acceptance scale for each of the nine items.** The semantic differential scale runs from −2 to 2. The figure also shows the $p$ values and effect sizes of a Wilcoxon signed-rank test comparing the Backup condition with the Forced condition per item.

condition can be explained because automated steering was enabled for the majority (76%) of the driving time.

The fact that the Forced condition yielded lower meanALE but higher meanMALE than the Manual condition indicates that a trade-off exists between automation use (i.e., more automation is better, as automation yields zero lateral error, thereby contributing to low meanALE) and automation reliability (i.e., if drivers are required to take over, as in the Forced condition, large performance errors can result). This cost-benefit trade-off resembles

the lumberjack effect, where automation has benefit for routine system performance, but a negative impact when the human has to take over (*Onnasch et al., 2014*).

Whether the driver is constantly in control of steering or whether he or she is occasionally forced to take control when looking away from the forward road, the maxALE did not obtain significant difference. An explanation for the observably large maxALE during the Forced condition could be that the steering wheel was not always centered during an automation-to-manual transition (see https://doi.org/10.4121/uuid:49d87edc-07a6-4f07-a5e6-0b699705881b for steering angle results). Whether or not a steering wheel should be decoupled during automated driving has been a topic of debate (*Kerschbaum, Lorenz & Bengler, 2014*). Our results suggest that a decoupled steering wheel is associated with increased lateral positioning variability if the decoupled steering wheel is not centered at the moment of transferring control back to the human driver.

## Workload

The Backup condition received the lowest self-reported workload ratings. During the Forced condition, drivers were monitoring what the automated pilot was doing, until they were forced back into control during periods of visual distraction. In other words, drivers initially experienced a state of low task demands and were forced into high task demands. This was not the case for the Backup condition, where adaptive automation was applied to help the human when task demands increased.

It should be noted that low workload ratings are not necessarily desirable, because low workload, or 'underload', may be associated with fatigue and loss of vigilance (*Hancock & Parasuraman, 1992*; *Young & Stanton, 2007*). *Parasuraman (2003)* argued that 'clumsy automation' can be an issue, whereby (adaptive) automation inadvertently adds workload (e.g., via new task demands like supervising or re-programming the automation) during already high periods of demand and do little to regulate workload during low periods of demands (i.e., during routine operation of the automation). In the end, an 'optimal' and balanced workload level should be aimed for. Within the current study, it is believed that the Backup condition supported such a balanced workload during driving because the presently 100% reliable automated steering did not require attention from the driver when it became active and it counteracted degraded lateral performance that would otherwise occur due to the uptake of the non-driving task.

## System acceptance

The Backup condition was rated more favorably than the Forced condition on nearly all items, except for the *Raising Alertness–Sleep-inducing item* where results were mixed and inconclusive. The task demands in the Forced condition provide an explanation of its negative acceptance ratings. Before the transition of control, participants experienced simultaneous demands to both monitor the automated driving and undertake the secondary task. Likewise, after the transition of control, participants were still involved in the secondary task when manual control was returned to them.

The results in Fig. 7 showed that participants looked away in higher proportions in the Backup condition than in the Manual condition. This suggests that the participants trusted

that the automation would assume control and were more inclined to keep their focus on the secondary task. One of the intended goals of the Forced condition was to prevent drivers from misusing driving automation that requires their active oversight and mental involvement (SAE Level 2 automation). However, the Forced condition appeared to show slightly more off-road glancing than the Manual condition. This is contrary to what was intended and expected with the Forced condition design as it was meant to return driver attention to the road. Apparently, in manual driving, participants are more conservative with their off-road glances than when automation is present (whether backup or forced). This may be because, in the former, there is one driving agent in the system whereas in the latter there are two driving agents.

When asked to complete a form at the end of the experiment, a majority of participants (22 of 31) preferred the Backup condition, which supports the results from the acceptance scale. These preferences add to the promise of the Backup condition in real-world applications. However, these preferences might also be because the automation lasts for as long as the driver keeps the eyes off the road, and so allows for unrestricted secondary task engagement. A driving simulator study by *Jamson et al. (2013)* found results which suggested that ''drivers are happy to forgo their supervisory responsibilities in preference of a more entertaining highly-automated drive'', whereas a test-track study by *Llaneras, Salinger & Green (2013)* showed that, when using reliable automation, drivers are likely to increase the frequency of secondary task interactions and engage in tasks that cause extended glances away from the road. In a review by *De Winter et al. (2014)*, it was found that relative to manual driving (100%), highly automated driving resulted in 261% of the number of tasks completed on an in-vehicle display. These findings suggest that the Backup condition might be preferred because it has the potential (whether intended or not by designers) to allow for increased end-user involvement in non-driving tasks.

## Limitations and generalizability issues
### Driving task simplicity

The track that the participants experienced was designed to be short-lasting (350 s per drive) and easy: no obstacles, other road users, or emergency situations were implemented. Furthermore, participants were instructed to keep the center of the lane and there was also no active penalty involved with an unintended lane crossing or large lateral position errors, and there was a reward for performing the secondary task well (in the form of a post-trial feedback score which was determined by the experimenter while the participant was performing the task). These factors may have caused participants to focus on the secondary task more than they would do in real life. Future research should establish how the adaptive automation would function in more naturalistic driving conditions.

### Eye-tracker capabilities

The eye tracker sometimes lost sight of the eyes of the driver and thus reported a null value for the gaze direction. The tracker appeared to have more difficulty with some drivers when compared to others. For our research we used a simple binary criterion to assess visual distraction: does the participant look at the monitor or not? This criterion was combined with a filter of 1.5 and 4.5 s interval (see 'Experimental conditions'), which accounted

for short data gaps due to e.g., blinking. Based on the results in Fig. 4, sensitivity of the on-monitor attention algorithms must have been high, as the percentage of participants for whom the automation was 'on' in the Backup condition was mostly zero when participants were supposed to look at the road (i.e., in between the secondary task periods). There were a few participants for whom the automation turned on during such periods in the Backup condition; we were unable to determine whether these were due to data losses of the eye-tracker or whether participants were actually looking away from the screen (e.g., exploring whether the Backup system was working properly). Specificity must also be high because it would be unlikely for the eye tracker to measure that a participant is looking at the monitor (which subtends a relatively small angular area in the participant's field of view) when he/she is looking instead at the CD-player. In summary, there were a few unexpected control transitions in between the secondary task periods, but these were infrequent and probably did not have a significant influence on the performance results.

The eye tracker used during this study had to be calibrated for every participant and sometimes still had trouble discerning the correct gaze direction. If the eye tracker were to calibrate itself and become more sensitive to gaze direction and less sensitive to confounding factors such as ambient lighting, this would increase the possibilities for real-world applications. Similar conclusions were drawn by *Pohl, Birk & Westervall (2007)* who also performed a study on distractions leading to lane departures.

### Realism of the steering wheel

The steering wheel that was used was smaller than an actual steering wheel and was designed without any force feedback. Some participants mentioned that this lack of force feedback was annoying. Another comment some of the participants made was that it was difficult for them to follow the instructions, which stated to completely focus on what they were doing with their hands. They were told that they were not allowed to perform any part of the secondary task blindly, to prevent a situation where they could just keep looking at the monitor and still finish the secondary task in time. This, understandably, might have felt unrealistic for the task of switching a CD in the CD-player (i.e., a task people may have sufficient practice with, and which could in principle be completed without continuous visual attention). Nonetheless, this approach was implemented to ensure that control transitions did take place and to simulate situations where long consecutive eyes-off-road periods did occur.

### Capabilities of the distraction detection algorithm

The initiation and termination threshold criteria for automatic transitions of control between the human and the automated driving were established based on pilot studies. However, these times are not necessarily generalizable, and would have to be determined again for experiments that use a different setup. For example, some drivers kept looking back and forth between the secondary task and the primary task at a high frequency. Due to this behavior, the algorithms never counted enough samples of looking away from the monitor which prevented the system from automatically initiating a control transition. A follow-up experiment could focus on discovering recommended initiation and termination times, or perhaps even incorporate an algorithm for using variable times.

The difference between safe and unsafe glances was defined by looking at the monitor or away from the monitor, respectively. In real-world driving situations, this would have to be defined more clearly. For example, further experiments might focus on what is considered as a safety region in the visual field. Perhaps it might be better to define a gradient where looking at the road directly in front of the car is or at task-relevant objects is considered to be 100% safe, whereas looking to the sides is less safe. Using such a gradient, the amount of time after which automation engages might also be varied so that, for example, a '10% safety area' uses a shorter initiation time than an '80% safety area'. It should also be noted that no mirrors were used during this experiment. Drivers usually look at the mirrors, and an improved algorithm should not classify mirror usage as a visual distraction.

Definitions of driver distraction (see *Pettitt, Burnett & Stevens, 2005*) are important for reliable driver monitoring and cross-study comparisons. Driver distraction can be separately categorized as visual, auditory, biomechanical, and cognitive (*Ranney et al., 2000*). It should be noted that the Backup and Forced systems detected visual distraction, not other types of distraction. For example, cognitive distraction is regarded as an important contributor to crashes, yet is a concept that is hard to define (*Young, 2012*). Cognitive distraction in driving (*Strayer et al., 2013*) has been discussed in different guises, including daydreaming (*Galéra et al., 2012*), mind wandering (*Yanko & Spalek, 2013*), looked-but-failed-to-see errors (*Sabey & Staughton, 1975*; *Staughton & Storie, 1977*; *Labbett & Langham, 2006*), cognitive tunneling (*Reimer, 2009*), attention focusing (*Chapman & Underwood, 1998*), loss of covert/peripheral attention via diminished functional field of view (*Crundall, Underwood & Chapman, 1999*), and highway hypnosis (*Wertheim, 1978*). We reiterate here that our Backup and Forced concepts cannot detect all forms of driver aberration: in reality, drivers may drive in an unsafe manner or crash into objects even when their eyes are on the road (*Victor et al., 2018*), and one should therefore not expect that the present Backup automation is a remedy to all types of driver distraction. However, given the predominant importance of visual information for driving (*Sivak, 1996*), the generally presumed eye-mind hypothesis where gaze direction is a strong correlate of cognitive activity (*Just & Carpenter, 1980*), and a substantial history of driving visual occlusion research (e.g., *Senders et al., 1967*; *Van der Horst, 2004*), adaptive automation based on visual attention alone could reasonably be expected to offer a beneficial contribution.

### Realism of the secondary task

The secondary task of changing a CD during this study was chosen because it was assumed to involve similar visual-manual loads as a number of common and risky in-vehicle tasks (e.g., texting, reaching for a dropped object, searching within a bag or purse, handling cables of charging devices, etc.). Participants were periodically forced to perform this secondary task at pre-defined moments during driving. This might have felt unnatural to some of the drivers because normally, a driver might choose a moment during driving before he or she would start a secondary task, whereas during this study these moments were forced.

### Mode errors and human machine interface

Because of the automatic and dynamic switching of driving task responsibility between the driver and the automated driving system, the Backup and Forced conditions could be susceptible to mode confusions, a well-known problem in human-automation interaction (e.g., *Feldhütter, Segler & Bengler, 2017*; *Sarter & Woods, 1995*). A mode confusion occurs when the driver believes that the automation is on while it is off, or vice versa (see *Janssen et al., in press* for a framework of mode confusions in automated driving).

In our study, the status of the automation was communicated visually to the driver by means of a status bar in the middle of the dashboard. However, because the secondary task imposed a visual distraction, it was difficult for the driver to know whether the automation had taken control or not, as predicted by the multiple resource theory (*Wickens, 2002*). In more complex driving tasks, where the driver performs many head movements (e.g., looking over the shoulder, looking in mirrors), the driver may be susceptible to mode confusion, as such conditions could cause the Backup automation to enable itself without the driver being aware of this.

A proper human-machine interface is essential to prevent such confusions and facilitate trust in the adaptive system. *Donmez et al. (2006)* found that display modality of a distraction-mitigation feedback system had a strong effect on driver acceptance and trust. Future research could be focused on how to best communicate the automation status to a visually distracted driver and whether the existence of backup automation needs to be communicated at all. For example, if the automated driving functions are implemented in an innocuous manner (e.g., small accelerations, minor corrections, blended inputs, etc.), automation status might even be best hidden to avoid confusion or misuse. That is, perhaps the driver does not need to know that the automation exists at all or when it is functioning (cf. electronic stability control, emergency enhanced braking power, etc.).

## CONCLUSIONS AND RECOMMENDATIONS

In conclusion, the Backup condition shows the potential to increase safety when compared to manual driving. A system that forces manual control back upon the driver appeared to be less safe than normal manual driving and less accepted than a backup system.

The current systems were designed to be simple and will need to be tested in more realistic long-lasting studies before any definitive conclusions can be drawn about the safety implications during real-world driving, and see *Kalra & Paddock (2016)* for calculations indicating that hundreds of millions of kilometers need to be driven in order to prove that automated driving technology is safe. Further testing might focus on expanding the simulation and the algorithm to account for other traffic, objects, emergency situations and increase fidelity by including car mirrors, and a more realistic car interior.

Finally, we note that the Backup and Forced conditions rest on different philosophies. That is, the Backup automation is a form of background automation (*Kyriakidis et al., in press*), where automation is engaged only when the driver is measured to be distracted. The assumption here is that, even though the automated driving system may be imperfect, automation is still better than a visually impaired human. The Forced automation system is

a form of foreground automation, where the automation is active for most of the time but needs a human supervisor at all times. In the Backup condition, participants could devote themselves more to the secondary task than in the Forced condition. This difference result could be interpreted as good (because a given secondary task is completed sooner) or bad (because it affords the ability to devote attention to the secondary task), depending on the context of operations.

It may take many decades of technological progress until fully automated (i.e., autonomous) driving is commercially viable (*Shladover, 2016*). Until that time, foreground and background automation strategies are viable candidates to be further researched developed before wide-market deployment on public roads.

### Funding

The research in this paper was conducted under the project HFAuto—Human Factors of Automated Driving (PITN-GA-2013-605817). The funders had no role in study design, data collection and analysis, decision to publish, or preparation of the manuscript.

### Grant Disclosures

The following grant information was disclosed by the authors:
HFAuto—Human Factors of Automated Driving: PITN-GA-2013-605817.

### Competing Interests

The authors declare there are no competing interests.

### Author Contributions

- Christopher D.D. Cabrall and Nico M. Janssen conceived and designed the experiments, performed the experiments, analyzed the data, contributed reagents/materials/analysis tools, prepared figures and/or tables, performed the computation work, authored or reviewed drafts of the paper, approved the final draft.
- Joost C.F. de Winter conceived and designed the experiments, analyzed the data, contributed reagents/materials/analysis tools, prepared figures and/or tables, performed the computation work, authored or reviewed drafts of the paper, approved the final draft.

### Ethics

The following information was supplied relating to ethical approvals (i.e., approving body and any reference numbers):

This research was approved by the Human Research Ethics Committee (HREC) of the Delft University of Technology (TU Delft).

### Data Availability

4TU.Centre for Research Data: https://doi.org/10.4121/uuid:49d87edc-07a6-4f07-a5e6-0b699705881b.

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
