# Peer review of "Adaptive automation: automatically (dis)engaging automation during visually distracted driving"

_PeerJ Computer Science, doi:10.7717/peerj-cs.166_

## Round 0.1 · original submission · Minor Revisions

Congratulations on the acceptance of your submission, subject to minor revisions! Please note that it is particularly important that you make available the supplementary material as specified by the reviewers.

·

Basic reporting

Well written.
Only a few small grammatical comments.
1. line 160 missing a period before "During"
2. line 229-231 "Furthermore, the participant was not allowed to perform the secondary task without looking at the secondary task, to ensure that he/she was not visually distracted when performing the secondary task." I find this sentence confusing, the first part seems to convey that they had to look at the CD player to complete the task, but the second part conveys that they were not supposed to be looking away while completing the secondary task.

Figure 4: Is this "proportion of participants with automation turned on" or "mean proportion of manual driving time"? The first, as it is described in the first sentence of the figure discussion, leads me to believe that the y axis should be proportion of participants. Also, the “automation status of individual participants”, is that indicating when automation was activated (for backup) and deactivated (for forced)?

Figure 5: Why is the magenta line vertically shorter in the forced condition? Does that convey a different experimental condition?

Figure 7: I think "percentage of participants glancing ON-ROAD" (or "OFF-ROAD") would be more insightful than "ON-SCREEN", as this makes it more intuitive which direction on the y axis is safer. Also, on-screen makes it less generalizable, as a screen is usually equated to an in-vehicle screen (e.g., infotainment system).

Figure 10: What is the y-axis? Are these raw scores or transformed? It doesn't really convey what value would be considered "better" as it currently stands.

Experimental design

I found the description of the automation very confusing, and only understood once I read the entire manuscript. This needs to be more clearly described up front, all at once. For example, I think the paragraph in line 576 should be incorporated with line 112, where you discuss the rational for using the "Forced condition". As it is currently written, the forced conditions seems like an unideal system that consumers would not want in their vehicle and that could yield liability issues to manufacturers. I also think it needs to be more clearly discussed that this is essentially cruise control or ACC with a lane keeping system, and that the lane keeping system is being manipulated. The "manual" condition is not entirely manual, as the vehicle is still cruising at 70 kph, correct? It's fine to use the term "manual", but I just think this needs to be more clearly defined - again perhaps earlier on, maybe in the abstract too? This would also help make the results more generalizable, because essentially you are looking at a partial automation condition, with only lane keeping engaged/not engaged, but always an ACC component.

Validity of the findings

line 368: “In summary, the more automation was applied, the better the overall average lane keeping performance.” I’m not convinced that “forced” can be described as more automation, it seems that if the system is forcing the driver to resume lateral control when they need it most (i.e., distracted), then it is not increasing automation.

Sections 4.1-4.3 doesn't really add much discussion, it is just re-summarizing the same information presented directly above. Perhaps incorporate more references to the literature, or discussion of interpretations, and this could be more meaningful.

Section 4.5 doesn't really discuss the generalizability of the findings, but more so is the first time the "forced" condition is rationalized for the experimental design. I think more expansion on how these findings could be used would be more fitting.

Additional comments

This is certainly an interesting topic, with a well thought experimental design and appropriate analysis. I think further clarity up front of the design would be helpful, as well as more consideration on how to best present the data in the figures.

Line 95-97: Usually include page number of reference in citation when you use a direct quote.

·

Basic reporting

This paper investigates how a car’s lane deviation is affected by the automation mode. Three situations are compared: manual driving, a car where the automation is the “back-up” that takes over when there is too much distraction (as determined based on eye-tracking), and a system that occasionally forces the human to drive (even when distracted).

The article meets most of PeerJ’s basic reporting criteria. The exceptions are:
1. The article occasionally refers to supplementary files for scripts and additional analyses, yet I cannot see these files in the online submission system. For a couple of claims in the manuscript, the details are essential.
2. I also cannot find files that contain the raw data, which seems to be a requirement of the journal.

I also have some more detailed feedback below to improve basic reporting.

Experimental design

The research question is well defined and is relevant, in the sense that it provides data-driven insights into the success of a “backup” and “forced” automated car system. The investigation is rigorous (looking at various aspects of lane deviation, gaze, and subjective experience).

For the Methods section, I would encourage the authors to improve on the following:
3. In section 2.5 the three conditions are explained. The initial explanation is quite high level, but later (line 200) some nuances are brought to the attention of the reader. I would encourage the authors to provide the high-level description outside of the methods section to the introduction (i.e., to introduce the general concept), and to focus in the methods section on the *exact* implementation of the method (i.e., the exact method that was used; here the details matter as the method is a specific implementation of the general idea). That reduces the chances that readers might overlook these important details.

Again there are also some more detailed comments on this item below.

Validity of the findings

In general, this meets the criteria of the journal, with some detailed comments below to further improve this.

Additional comments

Here are the more detailed comments that I hope can help to improve the manuscript and line of work. I ranked them in what, from my perspective, is relatively more “major” and more “minor”.

Relatively more major:
4. A hidden assumption behind the work is that gaze at the road (or screen) is indicative of attention for the road. This is a very strong assumption (by the way: which many other researchers also make either implicitly or explicitly). Given that gaze is the input for your system, it is important that some nuance is introduced in the paper. Specifically, I would encourage the authors to make more explicit that this is an assumption, and that there can be exceptions to the rule “where you gaze at is where you attend”. In general, the literature on “covert attention” can be useful in this regard.
To state a couple of examples of why the distinction between covert and overt attention is relevant: people with “tunnel vision” might look at the road, but not fully process all relevant information. Similarly, people who are mind wandering, or mentally distracted might gaze somewhere but not really attend to (or more neutral: Process) information at the gaze location.
It would be good to provide some nuance on this topic in the introduction (e.g., around line 69) and the general discussion.

5. The secondary task is introduced as a task that “commonly occurs while driving” (e.g., line 146). This claim is too strong — at least when concerning the specific implementation of the CD change that is used here, where (a) a lid needs to be opened, and (b) a physical stack of CD cases is handled. In particular, I am not aware of in-car CD players that adhere to property (a) above.
More generally, one might wonder whether modern cars are not more focused on MP3s and digital devices (e.g., use of phone connected to car) to play music, instead of CDs.
To avoid making too strong a claim, I think the better claim is that the authors used this task because it involves visual-manual distraction, and because the steps behind the task are detailed reported in another study (cited in line 218). That is, there is a relevant benchmark/reference for it.

6. Line 235: the participant was scored. It does not say explicitly who scored the participant. Was this an automated algorithm or the experiment leader? In general, the details (which should be in supplementary material, but I can’t find them) are needed here to assess whether this metric is valid, as (without knowing more details) it seems to involve both objective and subjective criteria. For the potential subjective component, some better understanding of the standards and training of the marker/experimenter are needed to interpret correctly. (or at the least: an acknowledgment that there is a subjective element to this — as I understand that this might be hard to report in hindsight)

7. Figure 4 now reports a score between 0 and 1, whereas this axis uses binary data (did a participant look yes/no) and then uses that data to assess what percentage or fraction of participants looks at the screen. The y-axis should, therefore, be changed to reflect percentage (or fraction) of users.

8. Line 518: these were “infrequent” and “never appeared to last long enough”. What was the rough criteria to call something “infrequent’ (e.g. “Less than X% of the cases) and what was the rough criterium to call something not “long enough”?


More minor:

9. Counterbalancing: On line 159 it is claimed that the study is counterbalanced. However, there are 31 participants and 3 conditions, and therefore 6 unique orderings (3 x 2 x 1). 31 participants does not allow for a fully counterbalanced design (i.e., that would require a multiple of 6, such as 30), and this is also not claimed by the authors. However, I also do not see in what other way this odd number can be balanced (wouldn’t a balanced design not require an even number?).
In other words: the exact way of counterbalancing should be explained more clearly.

10. Line 178 “inferences from the literature”: can the authors describe more precisely what the claims or data in the literature was and how this lead to the inference? For example, what is the range of values used or mentioned in the literature? And how does this range relate to your criteria?

11. Line 264: “If the calibration was successful”: I assume that this was determined by the eye-tracker system. What are the criteria for calling it successful?

12. This leads to a more general question: what is the accuracy of the eye-tracker system. Note that this can be expressed in terms of what the company defines (which are often relatively ideal values), and in terms of what you found (where the authors already mention some limitations). Do the authors have data on how often there is a “miss” or “False alarm” in terms of whether the participant is looking at the screen?

13. A more personal aspect, that does not need addressing in the paper, but which I hope is of interest to the authors: when I read the paper another hidden assumption seems to be that eventual implementation of such a system simply ‘works’ and that the human driver “knows” when it works. In particular, for the backup system, to function well the system really needs to work well, and the human must be able to rely on it. In practice, I suspect that such a system could lead to “mode confusion” (because maybe it can be turned off, and the human might misinterpret when it is on or off), and if a human incorrectly relies on it, or if the system does not function properly when it should, accidents might happen. I am reminded of this, as such mode confusion (and how to think about it systematically) is something I recently wrote about myself, and that might be of interest to the authors. See Janssen, Boyle, Kun, Ju, & Chuang (in press, International Journal of Human-Computer Interaction) (a preprint is on my website)


Signed,
Christian P. Janssen

---

## Round 0.2 · accepted · Accept

Thank you for your attention to the initial reviews!